# Automatic structure-based NMR methyl resonance assignment in large proteins

Iva Pritišanac [1], Julia M. Würz[1], T. Reid Alderson [2] & Peter Güntert [1,3,4]*

Isotopically labeled methyl groups provide NMR probes in large, otherwise deuterated proteins. However, the resonance assignment constitutes a bottleneck for broader applicability of methyl-based NMR. Here, we present the automated MethylFLYA method for the assignment of methyl groups that is based on methyl-methyl nuclear Overhauser effect spectroscopy (NOESY) peak lists. MethylFLYA is applied to five proteins (28–358 kDa) comprising a total of 708 isotope-labeled methyl groups, of which 612 contribute NOESY cross peaks. MethylFLYA confidently assigns 488 methyl groups, i.e. 80% of those with NOESY data. Of these, 459 agree with the reference, 6 were different, and 23 were without reference assignment. MethylFLYA assigns significantly more methyl groups than alternative algorithms, has an average error rate of 1%, modest runtimes of 0.4–1.2 h, and can handle arbitrary isotope labeling patterns and data from other types of NMR spectra.

[1] Institute of Biophysical Chemistry, Center for Biomolecular Magnetic Resonance, Goethe University Frankfurt am Main, Max-von-Laue-Str. 9, 60438 Frankfurt am Main, Germany. [2] Laboratory of Chemical Physics, NIDDK, National Institutes of Health, Bethesda, MD 20892-0520, USA. [3] Laboratory of Physical Chemistry, ETH Zürich, Vladimir-Prelog-Weg 2, 8093 Zürich, Switzerland. [4] Department of Chemistry, Tokyo Metropolitan University, 1-1 Minami-Osawa, Hachioji, Tokyo 192-0397, Japan. *email: guentert@em.uni-frankfurt.de

The last decade of structural biology has seen growing interest in biologically relevant large protein assemblies, as witnessed by an explosion of high- and low-resolution structural studies of macromolecular machines[1]. NMR spectroscopy is the principal experimental method for the simultaneous analysis of both the structures and dynamics of biomolecules at atomic resolution. The traditional size-limit of solution-state NMR spectroscopy, typically placed below 30 kDa, was overcome by Transverse Relaxation-Optimized SpectroscopY (TROSY)[2]. The TROSY enhancement, initially established for amide groups, was subsequently also realized for selectively methyl-labeled proteins (methyl-TROSY)[3,4]. Methyl-TROSY has since enabled studies of protein complexes in excess of 1 MDa[5] in unprecedented detail, revealing the mechanisms of dynamic molecular machines[6–8].

For optimal gains in the signal enhancement and resolution of methyl-TROSY spectra, selectively protonated, $^{13}$C-labelled methyl groups are introduced into an otherwise highly deuterated background[9]. To this end, cost-effective and robust biosynthetic strategies have been established for the selective or simultaneous labelling of all methyl-containing amino acids in *Escherichia coli*[10,11]. Selective labeling of methyl groups is also possible in eukaryotic expression systems[12–14]. The labeled methyl groups have favorable spectroscopic properties that render them observable also in large proteins and protein assemblies. Methyl groups are effective site-specific probes of molecular dynamics, structure, and interactions, as they are found both throughout the hydrophobic core of a protein and on its surface[10,15].

The major bottleneck for NMR studies with selective methyl-labeled proteins is the resonance assignment, i.e. relating $^1$H/$^{13}$C signals in the NMR spectra to specific methyl groups in the protein (Fig. 1)[16]. In small and medium-size proteins, NMR signals from the protein backbone can be observed and used in triple-resonance, "through-bond" experiments for the sequence-specific resonance assignment of the backbone[17], to which side-chain methyl resonances can be linked[18]. In contrast, for large proteins, backbone resonances and triple-resonance spectra cannot be observed, and, unless the protein is modified, only nuclear Overhauser effects (NOEs) between methyl groups remain accessible as NMR input data for assignment.

Assignment strategies for large proteins or proteins assemblies include divide-and-conquer approaches wherein sufficiently small individual protein domains or subunits are produced separately, such that their backbone resonance assignment can be determined using standard methods[19]. This approach requires that the resonance frequencies of the subsystems coincide closely with those of the complete system. To complete the assignment, the approach is often supplemented with site-directed mutagenesis of individual methyl-bearing residues[20,21]. As an alternative, a high-resolution structure of the studied protein or complex can be utilized in combination with NMR experiments that reveal spatial proximity between methyl groups[22,23], or between methyls and site-specifically attached paramagnetic probes[24].

The laborious and time-consuming nature of these assignment strategies prompted automation efforts. Presently, two groups of structure-based, automatic assignment approaches are available: NOE spectroscopy (NOESY) and paramagnetism-based methods. Both rely on NMR-derived, sparse distance measurements that are compared to a known three-dimensional (3D) structure. Paramagnetic approaches require the site-specific introduction of paramagnetic probes and estimates of the magnetic susceptibility tensors. These approaches define the optimal methyl assignments as those that minimize the difference between the measured and the calculated paramagnetic observables[25–27]. For instance, PRE-ASSIGN[27] uses paramagnetic relaxation enhancements (PREs),

whereas PARAssign[26] relies on pseudo-contact shifts (PCSs). NOESY-based automatic approaches match a network of measured methyl–methyl NOE contacts to the network of short inter-methyl distances predicted from the protein structure, using Monte Carlo[28–31] or graph-based[32,33] algorithms. For example, MAGMA[32] uses exact graph matching algorithms to generate confident assignments for a subset of well inter-connected methyls. For the remaining methyls, MAGMA reports all ambiguous assignment possibilities, which may be used for further experimental investigation.

Automated methods for structure-based methyl resonance assignments can be characterized by the experimental requirements for measuring the input data, and by the completeness and accuracy of the assignments that they produce. An optimal algorithm functions with data that can be measured readily, tolerates experimental imperfections, is computationally efficient, and yields confident assignments for a large fraction of all methyls. To minimize the amount of error or subsequent manual checking, the algorithm (not the user) should distinguish confident assignments, which are almost certainly correct, from other, tentative or ambiguous ones. Existing algorithms fall short of this ideal in different ways.

Therefore, we here adopt the FuLlY Automated assignment algorithm FLYA[34], (Fig. 1) which is integrated in the CYANA structure calculation software[35] and has been shown capable to assign proteins exclusively from NOESY data[36], for structure- and NOESY-based methyl resonance assignment. We apply the resulting MethylFLYA algorithm to a benchmark[32] of five large proteins and protein complexes and show that, on the basis of methyl–methyl NOEs alone, MethylFLYA can assign significantly more methyl resonances with high accuracy than the previously introduced methods MAGMA[32], MAP-XSII[29], FLAMEnGO2.0[31], and MAGIC[33] operating on the same input data. To demonstrate its robustness with respect to ambiguous and imperfect experimental information, we apply MethylFLYA also to unrefined peak lists, reduced input data sets, and peak lists obtained by automated peak picking with the CYPICK algorithm[37].

## Results

**Benchmark data**. MethylFLYA was applied to the five largest proteins of a benchmark data set that was originally prepared for evaluating the MAGMA algorithm for automated methyl assignment, as described in the original publication[32]. In addition, methyl NMR data for the 20 kDa N-terminal domain of Heat Shock Protein 90 (called HSP90 in this paper)[38], which has also been used previously with MAGMA, were used for evaluating MethylFLYA in combination with automated peak picking with CYPICK[37]. The main benchmark data set comprised five proteins of varying molecular mass and shape for which NOESY data from specifically methyl-labeled samples, assignments, and 3D structures are available (Table 1):[32] the N-terminal domain of *E. coli* Enzyme I (called EIN in this paper; molecular mass 28 kDa)[39], a dimer of regulatory chains of aspartate trans-carbamoylase from *E. coli* (ATCase; 34 kDa)[24], maltose binding protein (MBP; 41 kDa)[40], malate synthase G (MSG; 81 kDa)[15,18], and the "half-proteasome" 20S core particle, a 14-mer ($\alpha_7\alpha_7$; 358 kDa)[41].

The following experimental data were taken from the MAGMA benchmark[32]: (i) Assigned [$^1$H,$^{13}$C]-HMQC peak lists providing reference assignments, which were used only to evaluate the accuracy of the MethylFLYA results, while unassigned versions of these [$^1$H,$^{13}$C]-HMQC peak lists were supplied to MethylFLYA. (ii) Filtered and unfiltered (see below) NOESY peak lists from 3D (ATCase, $\alpha_7\alpha_7$) or 4D (EIN, MBP, MSG) methyl–methyl NOESY spectra. (iii) Solution or crystal structures of the proteins, taken

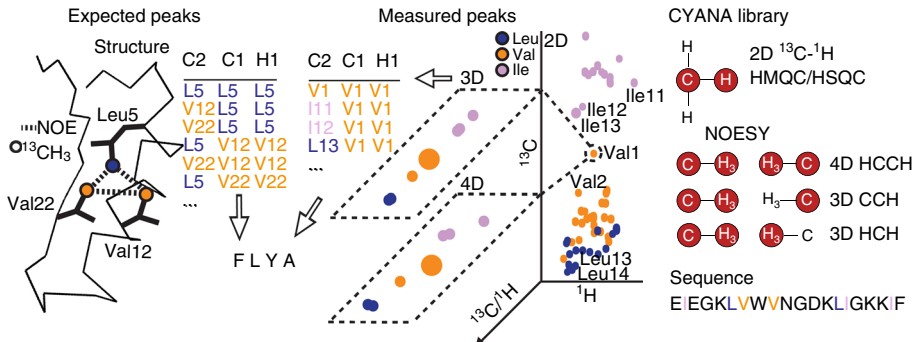

**Fig. 1** Automatic methyl resonance assignment with MethylFLYA. The expected methyl–methyl contacts are computed from a 3D structure of the protein (left). The contacts are written to a list of expected NOE peaks with the amino acid type of each contact indicated (e.g. I–V, L–L). A list of measured NOESY peaks is obtained by manual (or automatic) inspection of the 3D or 4D methyl–methyl NOESY spectrum (center) guided by information from the 2D [$^1$H,$^{13}$C]-HMQC spectrum. If known, the amino acid types of the methyl peaks that give rise to measured NOEs are included in the peak list. The [$^1$H,$^{13}$C]-HMQC peak list and the expected and measured NOE peak lists are supplied to MethylFLYA for the automatic methyl assignment calculation. In addition to the peak lists, the MethylFLYA calculation requires the protein sequence and knowledge of the magnetization transfer pathways for the employed NMR experiments, which are provided in the CYANA library (right)

| Table 1 Methyl resonance assignment statistics | | | | | |
|---|---|---|---|---|---|
| | **EIN** | **ATCase** | **MBP** | **MSG** | **α₇α₇** |
| Protein properties: | | | | | |
| Residues per monomer | 259 | 153 | 370 | 731 | 233 |
| Multimeric state | monomer | dimer | monomer | monomer | 14-mer |
| Molecular mass of multimer (kDa) | 28.3 | 34.2 | 40.6 | 81.4 | 358.4 |
| Experimental NMR data: | | | | | |
| Labeled amino acids | AILV | ILV | ILV | ILV | ILV |
| NOESY dimensions | HCCH | CCH | HCCH | HCCH | CCH |
| Labeled methyl $^1$H-$^{13}$C groups: | | | | | |
| All | 146 | 66 | 123 | 276 | 97 |
| With reference assignment | 133 | 62 | 123 | 268 | 93 |
| With NOESY peaks | 116 | 54 | 118 | 236 | 88 |
| Methyl resonances confidently assigned by MethylFLYA: | | | | | |
| All[a] | 101 | 45 | 86 | 176 | 80 |
| Correct | 90 | 35 | 83 | 173 | 78 |
| Erroneous | 0 | 4 | 2 | 0 | 0 |
| Assigned by MethylFLYA using unfiltered peak lists: | | | | | |
| All[a] | 107 | 42 | 89 | 183 | 82 |
| Correct | 98 | 35 | 87 | 174 | 80 |
| Erroneous | 0 | 3 | 2 | 3 | 0 |
| Assigned by MethylFLYA using a single distance cutoff: | | | | | |
| All[a] | 118 | 44 | 96 | 194 | 84 |
| Correct | 101 | 35 | 85 | 176 | 80 |
| Erroneous | 4 | 3 | 8 | 8 | 1 |
| Methyl resonances assigned by MAGMA:[32] | | | | | |
| Correct | 56 | 18 | 78 | 97 | 84 |
| Erroneous | 0 | 0 | 0 | 2 | 0 |
| Methyl resonances assigned by MAP-XSII:[29] | | | | | |
| Correct | 64 | 24 | 16 | 33 | 79 |
| Erroneous | 17 | 5 | 11 | 9 | 1 |
| Methyl resonances assigned by FLAMEnGO2.0:[31] | | | | | |
| Correct | 0 | 8 | 35 | 0 | 70 |
| Erroneous | 0 | 4 | 0 | 0 | 18 |
| Methyl resonances assigned by MAGIC:[33] | | | | | |
| Correct | 71 | 39 | – | – | – |
| Erroneous | 13 | 6 | – | – | – |

Filtered input NOESY peak lists were used, unless noted otherwise. See text for details
[a]All strong assignments, including correct, erroneous, and additional ones for methyl groups without reference assignment

from the Protein Data Bank with accession codes 1EZA for EIN, 1D09 for ATCase, 1EZ9 for MBP, 1D8C for MSG, and 1YAU for α₇α₇. In addition, MethylFLYA calculations were performed for the alternative structural forms 1TUG for ATCase, 3MBP for MBP, and 1Y8B for MSG. Automated peak picking with CYPICK was performed for NOESY spectra in Sparky[42] format for EIN, ATCase, and HSP90. Information about Leu/Val geminal methyl pairs, which was available in the MAGMA benchmark[32], was incorporated into the MethylFLYA calculations in the form of simulated HCcCH TOCSY peak lists.

Two sets of experimental methyl–methyl NOESY peak lists were used for the five proteins. The first set ("filtered peak lists") comprised peak lists from the MAGMA study that were filtered for reciprocity of donor and acceptor NOE cross peaks (only the reciprocated peaks were kept), and signal-to-noise ratios (only the peaks with $S/N \geq 2$ were kept)[32]. The second set comprised unfiltered ("raw") peak lists, generated by manual analysis of NOESY spectra using Sparky[42] software, which were not manually modified before the assignment calculation.

**MethylFLYA parameter optimization**. While most parameters of the MethylFLYA algorithm could be kept at the values that had been found optimal in earlier applications of the original FLYA algorithm[34,36,43–46], specific optimization of a small number of parameters that are of relevance to structure-based methyl assignment was advantageous.

MethylFLYA considers only methyl–methyl distances shorter than a user-defined cutoff $d_{cut}$ for generating expected methyl–methyl NOESY cross peaks based on a protein structure (see Methods). In addition, each expected peak is attributed a probability value to (roughly) reflect the probability of observing it in the corresponding measured spectrum. For expected NOESY cross peaks, we tested a range of distance cutoffs and distance-dependent observation probabilities (Supplementary Fig. 1). Across these parameter values, we monitored the fraction of correct and incorrect strong (i.e. confident) methyl assignments and the percentage of explained input NMR data (methyl–methyl NOEs). Even though protein-specific profiles can be observed in Supplementary Fig. 1, the fractions of assigned methyl resonances generally plateaued around $d_{cut} = 5\,Å$ for EIN, ATCase, MBP, and MSG, or $d_{cut} = 6\,Å$ for $\alpha_7\alpha_7$ (Supplementary Fig. 1). These plateaus coincided with about 80% explained input NMR data, which was determined as optimal for these data sets. Increasing the observation probabilities generally diminished the quality of the results, as more incorrect assignments were obtained (Supplementary Fig. 2). Predictably, more of the observed NOEs were assigned using higher distance cutoffs for generating the expected NOEs, but assignment errors also increased. In most cases, the assignment accuracy peaked around the plateaus of assigned methyl fractions and decreased at higher ($\geq 7\,Å$) and lower ($\leq 4\,Å$) distance cutoffs. To reduce the dependency on small variations of the distance cutoff, we always performed three assignment calculations using the given $d_{cut}$ as well as a slightly lower and a slightly higher value, i.e. $d_{cut} - 0.5\,Å$, $d_{cut}$, and $d_{cut} + 0.5\,Å$, and we required assignments to be self-consistent over the three runs (see Methods). As such, based on Supplementary Figs. 1 and 2, we used $d_{cut}$ values of 4.5, 5.0, and 5.5 Å for EIN, ATCase, MBP, and MSG, and 5.5, 6.0, and 6.5 Å for $\alpha_7\alpha_7$, as well as a NOESY cross peak observation probability of 0.1 for all following MethylFLYA calculations.

The number of independent assignment optimization runs that is necessary for obtaining reproducible, virtually seed-independent strong assignments was also optimized (Supplementary Fig. 3). All further MethylFLYA runs comprised 100 independent assignment optimization runs.

**Assignment completeness and accuracy**. We evaluated the performance of MethylFLYA on manually (expert) picked methyl NOE signals that were either (i) filtered to keep only the NOESY cross peaks that are observed reciprocally between two methyl groups and that are above a defined signal-to-noise threshold[32] ("filtered" peak lists); or (ii) used without any subsequent editing or filtering ("unfiltered"/"raw" peak lists). Using manually analyzed and filtered NOE data (i)[32], MethylFLYA assigned between 63% (ATCase) and 84% ($\alpha_7\alpha_7$) of the methyl resonances for

which reference assignments are available (Fig. 2b; Table 1, Supplementary Table 1), with no assignment errors for EIN, MSG, and $\alpha_7\alpha_7$. Two incorrect methyl assignments were found for MBP, and four for ATCase (Fig. 2b). In the 3D structures, all incorrectly assigned methyls are located in proximity to their correct assignment positions (Supplementary Fig. 4, Supplementary Table 2). Such spatially localized assignment errors are expected to have minor impact on studies that do not require very high-resolution information, for instance, when identifying an interaction interface.

We also note that more stringent criteria can be applied to define the confident (strong) methyl assignments, which further reduce errors. For instance, increasing the requirement for self-consistency of assignments from multiple parallel runs of the algorithm from 80% to 90% (see Methods), results in a decrease in error for ATCase from 6% to 1%. This is achieved at the expense of reducing the percentage of strong assignments on average by 6%. It is thus possible to ensure a higher accuracy by "sacrificing" some of the strong assignments.

On the other hand, using a single distance cutoff (5 Å for EIN, ATCase, MBP, MSG; 6 Å for $\alpha_7\alpha_7$) instead of three cutoffs spaced by 0.5 Å for generating the expected NOESY cross peaks in MethylFLYA increases the overall number of assignment errors about four-fold (Table 1). It is thus not advisable even though the total number of strong assignments rises by about 10%.

Importantly, MethylFLYA is robust with respect to the presence of ambiguous or incorrect methyl–methyl NOEs, as judged by its comparable, or in some cases even better, performance on "raw" NOESY peak lists that were not filtered for NOE cross peak reciprocities and signal-to-noise ratios and retained any ambiguous and tentative NOE cross peaks (Fig. 2).

A spatial clustering of strong assignments can be discerned in the structures of EIN, ATCase, and MSG (Fig. 2c). This is likely due to the low number of long-range NOEs between the clusters. In addition to the strong assignments, MethylFLYA outputs ambiguous assignment options for all resonances to which at least one inter-methyl NOE is attributed. The number of ambiguous assignment possibilities to be displayed can be specified by the user.

**Reduced data sets**. We tested the performance of MethylFLYA on the benchmark when experimental information provided to the algorithm was reduced (Fig. 3). In the best-case scenario, both knowledge of the amino acid types of methyl resonances and linkage of the two geminal methyl groups of Leu and Val is available (Fig. 3a, c, black). The Ile-$\delta_1$ resonances are usually readily identified due to their upfield shifted $^{13}$C frequencies. To discriminate between Leu and Val resonances, separate protein samples can be prepared using selective labelling schemes. For instance, selective Leu labelling can be achieved by using $^{13}$C-labeled $\alpha$-ketoisocaproate[47], whereas the combined addition of unlabeled $\alpha$-ketoisocaproate and labeled $\alpha$-ketoisovalerate leads to exclusive labeling of Val[48]. To connect resonances from the two geminal Leu/Val methyl groups, an additional protein sample can be prepared in which both Leu/Val-methyl groups are protonated and $^{13}$C-labelled. A short-mixing time NOESY experiment can then be used to record cross peaks between geminal methyl groups[21,32] (Fig. 3a). Without discrimination between Leu and Val resonances, MethylFLYA performed very similarly as in the best-case scenario for EIN, MSG, and $\alpha_7\alpha_7$, confidently assigning 68, 62, and 79% of the methyl resonances, respectively, with complete accuracy (Fig. 3c, dark gray). For ATCase and MBP, the percentage of accurate confident assignments decreased by 3%. However, for ATCase the percentage of errors was also reduced simultaneously by 3%.

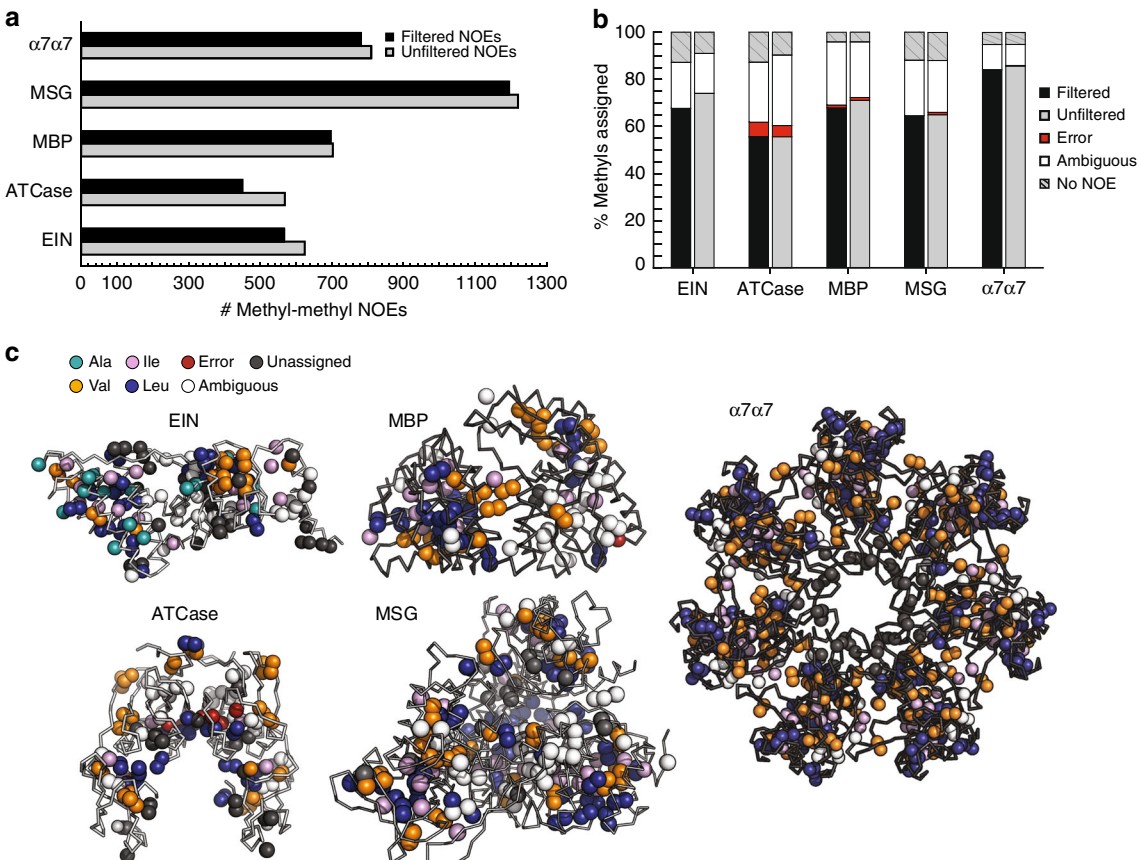

**Fig. 2** MethylFLYA performance on the benchmark. **a** Number of methyl–methyl NOEs before and after filtering of manually picked NOESY peak lists (see Results, Benchmark data). **b** MethylFLYA performance on filtered (black) and "raw" (unfiltered) manually picked NOESY peak lists (gray). **c** Methyl groups assigned as strong (confident) by MethylFLYA with filtered NOESY peak lists are indicated with colored spheres in the 3D structures of the benchmark proteins. The colors indicate the amino-acid types of the assigned groups, with non-assigned groups colored white

Removing the geminal Leu/Val pairing had a more significant impact, reducing the percentage of assigned methyls by ~19% for EIN, ATCase, MBP, and MSG, and up to 30% for $\alpha_7\alpha_7$ (Fig. 3c, light gray). The overall accuracy, however, remained high. The critical importance of this restraint for automatic methyl assignment was reported previously in the MAGMA study[32]. In the MAGIC study, a four-fold decrease in computational time and a somewhat improved assignment accuracy were noted as benefits of the restraint[33]. As an alternative, the information about Leu/Val geminal pairs can be substituted with stereospecific labelling schemes that restrict isotopic labeling to only pro-$R$ or pro-$S$ methyl groups, and thus reduce the number of methyl resonances in the [$^1$H,$^{13}$C]-HMQC spectrum[49]. For MethylFLYA, removing both the Leu/Val-geminal pairing and discrimination between Leu/Val methyl resonances led to a similar outcome as geminal pairing removal alone (Fig. 3c, silver), and led overall only to a slight further increase in erroneous assignments (1–2%). Interestingly, for ATCase, removing the Leu/Val resonance discrimination always improved the accuracy (Fig. 3c, dark gray, silver). We conclude that, especially for smaller proteins (<80 kDa), Leu/Val residue discrimination is not crucial for MethylFLYA.

The computation time of MethylFLYA scaled approximately linearly with the number of methyl groups in the protein. The complete protocol took between 0.36 and 1.53 h (Supplementary Table 3). Negligible differences in speed were noted for the calculations with lower input information content (Fig. 3, Supplementary Table 3). This illustrates the ability of MethylFLYA to efficiently deliver high-quality assignments even from considerably reduced input data.

**Combination with automated peak picking.** All currently available automatic methyl resonance assignment strategies rely, to different extents, on a manual analysis and interpretation of the NMR data. The NOE-based methods, for instance, require manual, expert, inspection of methyl–methyl NOESY spectra to generate peak lists as input to the assignment software[28–33]. We investigated whether an automatic peak picking algorithm, CYPICK[37], could be used in combination with MethylFLYA to fully automate methyl resonance assignment[50]. We tested the CYPICK-MethylFLYA combination on three proteins from the MAGMA study for which methyl–methyl NOESY spectra were available (Fig. 4). For these spectra, CYPICK found 77–83% of the manually identified methyl–methyl NOEs (Supplementary Fig. 5, Supplementary Table 4), which is comparable to its performance previously reported on 3D $^{13}$C-edited and $^{15}$N-edited NOESY spectra[37]. The somewhat high CYPICK artifact scores for EIN (34%) and HSP90 (46%) did not result in assignment errors, as only one methyl group misassignment was found for EIN and three for HSP90. Moreover, for EIN, even slightly more methyls were confidently and accurately assigned when the automatically generated CYPICK peak lists (78%) were used compared to the manually prepared lists (68%).

Despite the relatively large number of assignments for EIN, similar success was not found for the HSP90 and ATCase CYPICK datasets. In the case of HSP90, the considerably smaller

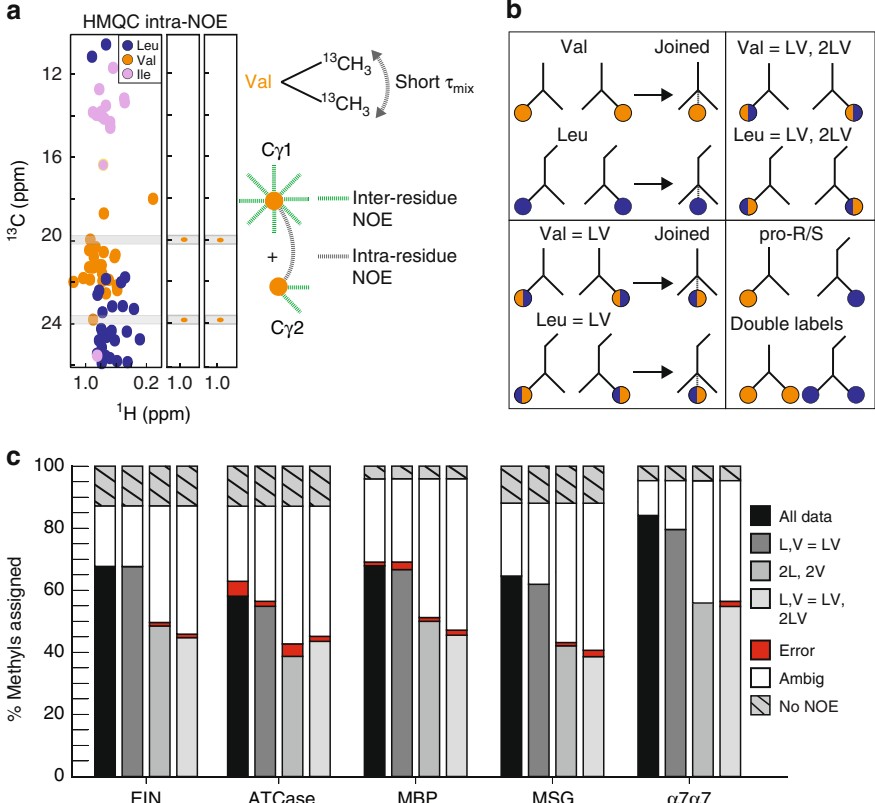

**Fig. 3** MethylFLYA performance with reduced data input. **a** Geminal methyl groups of Leu/Val residues can be linked with a short-mixing time NOESY experiment on an exclusively double methyl-labeled ([$^{13}\delta_1/^{13}\delta_2$]-Leu, [$^{13}\gamma_1/^{13}\gamma_2$]-Val) protein sample. In the NOESY plane of each Leu/Val methyl resonance, a strong signal from its geminal methyl resonance is observed (right). **b** Different possibilities for treating Leu/Val methyl resonances in MethylFLYA. Top, left: differentiation of the Leu- and Val-type of methyl resonances and known connectivity between geminal Leu or Val methyl groups. Bottom, left: no differentiation between methyl resonances of the Leu- or Val-type, but known connectivity between geminal methyl groups. Top, right: no differentiation between methyl resonances of the Leu/Val-type, nor knowledge of the geminal methyl connectivity. Bottom, right: stereospecific labelling of Leu/Val-methyl groups (pro-R or pro-S), or double labelling (both pro-R and pro-S). **c** MethylFLYA results with reduced data input. The percentage of assigned methyl groups given knowledge of all methyl amino acid types and the geminal Leu/Val-methyl connectivity is shown in black (all data). In dark gray (L, V = LV), knowledge of Ile- and Ala-methyl resonance types is assumed, but there is no discrimination between Leu or Val methyl resonance types. The geminal (Leu/Val) methyl resonances are connected. In light gray (2L, 2V), knowledge of all amino-acid types is assumed, but the geminal pairing of Leu/Val resonances is omitted. In silver (L, V = LV, 2LV), neither the amino-acid type nor the geminal pairing is known for Leu/Val methyl resonances. Knowledge of the amino acid types of other resonances (Ala, Ile) is still assumed

amount of assigned methyls could be attributed to the lower percentage of explained NOE data when using the CYPICK lists (Fig. 4b). When the manually generated NOE list was used, the MethylFLYA assignments explained roughly 80% of the NOE data at a 5 Å distance cutoff (Supplementary Fig. 5), consistent with the results presented above for the five proteins of the benchmark. In contrast, at the same distance cutoff, only about 60% of the NOE data were explained for the CYPICK-derived list. For ATCase, less than 40% of the methyl groups could be assigned, except for a single $d_{cut}$ value (Supplementary Fig. 5). The considerably worse performance of CYPICK-MethylFLYA on ATCase and HSP90 suggests that some methyl–methyl NOEs are more critical determinants of assignment success than the others. Overall, manual peak picking of the NOESY spectra (or manual screening and adaptation of automatically prepared peak lists) remains the best approach for preparing the input data for MethylFLYA.

**MethylFLYA using minimal input information.** All automatic methyl resonance assignment protocols that are presently available assume that the resonance frequency positions of $^1$H-$^{13}$C correlations from the 2D [$^1$H-$^{13}$C]-HMQC spectrum are known

and that each methyl resonance is associated unambiguously or, in some cases, ambiguously with a methyl residue type specified by the user. The benchmark data set of proteins from the MAGMA study[32], which was reused here, provides $^1$H-$^{13}$C resonance frequency positions based on the known reference assignment. However, the knowledge of these exact positions offers an additional source of information to the automatic assignment protocols, as it immediately resolves some inherent uncertainties, e.g., peak duplications or overlaps, and subsequently aids during the algorithmic attribution of NOEs to specific methyl-bearing residues.

Therefore, we sought to address the performance of Methyl-FLYA starting solely from 2D HMQC and 4D methyl–methyl NOESY spectra, whilst being 'blind' to the known $^1$H-$^{13}$C resonance frequencies and methyl residue types. To this end, we performed both manual and automated peak picking of both the 2D [$^1$H,$^{13}$C]-HMQC and 4D methyl–methyl NOESY spectra of EIN (see Methods, Supplementary Table 5, Supplementary Figs. 6, 7). The methyl residue types (Ala, Ile, and ambiguous Leu/Val), were assigned based on the BMRB chemical shift statistics[51] and the known number of expected peaks of different residue types (see Methods, Supplementary Fig. 7). The high degree of

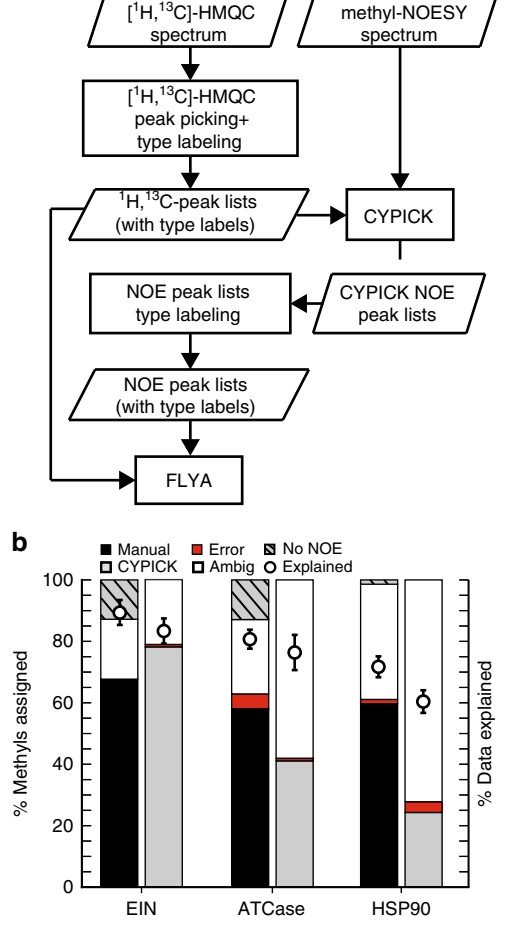

**Fig. 4** MethylFLYA performance with automatic peak picking using CYPICK. **a** An outline of the combined CYPICK-MethylFLYA assignment protocol (see Methods). "Type labeling"refers to the attribution of methyl resonances to amino-acid types (e.g. Ala, Ile, Leu, Val). **b** Comparison of MethylFLYA performance on manually and automatically picked NOESY data. The percentage of assigned methyl resonances and the percentage of explained input NOESY data, given the MethylFLYA assignment, are shown. The error-bars are standard deviations of the percentage of data explained over the three distance cutoffs (optimal cutoff ±0.5 Å, see Methods). Note that CYPICK does not assign methyl–methyl NOEs, and hence the identity of the methyl resonances with no NOEs ("no NOE") is not indicated for the CYPICK bars

overlap between the average methyl chemical shifts of Leu and Val rendered them difficult to separate, and therefore Methyl-FLYA treated them as ambiguous. The methyl peaks falling in overlapping regions for other residue types, i.e. Ala and Leu/Val, were either assigned a type based on the "best guess" (see Methods, Supplementary Fig. 7), or assigned ambiguously to the three possible types (Ala or ambiguous Leu/Val). As anticipated, when provided with the minimum amount of information, the percentage of strong (confident) assignments dropped significantly, from 74% attained when using the unfiltered methyl–methyl NOESY peak list, the known $^1H$-$^{13}C$ frequencies with known residue types, and the geminal Leu/Val resonance pairing (Fig. 2), to 30% or 24% when using only the manually or CYPICK-analyzed [$^1H$,$^{13}C$]-HMQC and 4D methyl–methyl NOESY spectra, respectively (Supplementary Fig. 7D). Nonetheless, the accuracy of the strong assignments remained high with only three and four methyls assigned incorrectly for

manually- and CYPICK-acquired peak lists, respectively (Supplementary Fig. 7E). The robustness of MethylFLYA to the highly ambiguous and partially incorrect input information is notable, especially when considering that the BMRB-derived assignment of residue types led to seven resonances with incorrectly assigned methyl residue type labels (Supplementary Fig. 7A, B). In fact, the assignment reflected this erroneous input information, as Ala12 and Ala160, that were attributed the wrong residue type, were assigned incorrectly by both approaches (Supplementary Fig. 7A, E). We here note that the user can use structure-based methyl resonance predictors, such as SHIFTX2[52], to flag the methyls that are expected to deviate significantly from BMRB chemical shift statistics and therefore likely to acquire an incorrect residue type label and subsequent misassignment (Supplementary Fig. 8). In addition, the user can exclude from consideration any strong assignments to the methyl resonances with a highly ambiguous residue type (i.e., Ala or ambiguous Leu/Val). If structure-based methyl resonance prediction is applied to exclude from consideration any strong assignments of the methyls that are expected to have a misassigned residue type (Supplementary Fig. 8), the assignment errors are reduced to only one or two methyls using manually or CYPICK-acquired peak lists, respectively (Supplementary Fig. 7E (i), (iii)). Attributing the methyl peaks in the overlapped regions of Ala/Val methyl residue types to both Ala and Leu/Val resulted in a somewhat higher percentage of strong assignments, but a further increase in errors (Supplementary Fig. 7D), and is therefore not recommended with the present implementation of MethylFLYA.

We next considered the performance of MethylFLYA when additional information in the form of the geminal Leu/Val methyl pairing is provided, albeit only for the well-resolved Leu/Val resonances in the spectrum (Supplementary Table 5, Supplementary Fig. 7C), as an unambiguous pairing of the geminal pairs is expected to be challenging or impossible for overlapping peaks. This assumes the preparation of an additional protein sample with both Leu/Val methyl groups labeled simultaneously. Next to providing an additional restraint for every pair of Leu/Val methyl resonances, such a sample would additionally distinguish unambiguously between Ala and Leu/Val types based on the 2D [$^1H$,$^{13}C$]-HMQC spectrum. Accordingly, in these calculations, the methyl residue type annotation was corrected for Ala and Leu/Val types. This resulted in the geminal pairing of 62 out of the 92 Leu/Val resonances expected based on the protein sequence. Introducing the additional information more than doubled the fraction of strong methyl assignments from 30% to 64% or 24% to 59% for manually or CYPICK-acquired peak lists, respectively. The errors mostly mapped to positions that are spatially proximal to the correct assignment (Supplementary Fig. 7E (ii), (iv)). We noted that the error in assignment of Val176γ$_2$ to Val156γ$_1$ occurred both when using manually- and CYPICK-picked peak lists, being the sole error in the latter case. The reference assignment showed that the methyl resonance of Val176γ$_2$ is overlapped with that of Leu123δ$_1$ in the 2D [$^1H$,$^{13}C$]-HMQC spectrum. As such, neither of the resonances were paired with their geminal methyls in the calculations. Resonance overlap is expected to prevent an unambiguous assignment of methyl–methyl NOEs using the automatic methyl NOESY assignment protocol of MethylFLYA, which, combined with the lack of any additional restraints for the resonances, such as the geminal methyl pairing, likely underlies the assignment error. This example illustrates how overlapping peaks can constitute a challenge for automatic methyl resonance assignment, which represents an important aspect for future improvement. Overall, the results provide a fair estimate of the lower bound of the performance of MethylFLYA given minimal data input and maximal data uncertainty, and demonstrate how additional

restraints introduced into the assignment search can significantly improve its outcome (Supplementary Table 5, Supplementary Fig. 7).

**Performance comparison.** The MAGMA study[32] included a performance comparison with the available NOE-based automatic methyl assignment software packages, MAP-XSII[29], and FLAMEnGO2.0[31]. For a comparison of the available methods, we used here the results for all proteins[32], apart from MSG, for which a different structure of the protein was used (Fig. 5, Supplementary Table 1, Supplementary Fig. 9). The recently introduced MAGIC[33] method requires the knowledge of signal intensities for all methyl–methyl NOE cross peaks, information that was not available for three out of the five proteins of our benchmark set: methyl–methyl NOESY spectra were available for EIN and ATCase, and in addition for HSP90. The performance of the MAGIC method on these datasets is summarized in Supplementary Table 6 and Supplementary Fig. 10.

## Discussion

Compared to the alternatives, MethylFLYA generated more confident and correct methyl assignments in all cases except for $\alpha_7\alpha_7$ (Table 1, Fig. 5), where all methods assigned more than 85% of the methyls. For the other proteins, MethylFLYA generated on average 18% more assignments than the next best performing software. Overall, MethylFLYA generated the highest number of confident and correct methyl $^1H$ and $^{13}C$ resonance assignments on this benchmark (confident and correct/total = 459/465), followed by MAGMA (333/335), MAP-XSII (216/259), and FLAMEnGO2.0 (113/135). Across the entire benchmark, MethylFLYA made assignment errors for six methyls. Based on the error rate on this benchmark, MethylFLYA is the second most accurate method after MAGMA, which made assignment errors only for two methyls. The latter two errors result from the use of a crystal structure for MSG (PDB 1D8C) instead of the NMR-derived structure (PDB 1Y8B) that had been used in the original MAGMA benchmark[32]. In the original study, MAGMA was reportedly sensitive to the structural difference between the two forms, which is likely due to the presence of the ligand in the crystal structure[32]. Here, we tested the performance of all methods exclusively on crystal structures to omit the need for NMR structures, which are anticipated to be unavailable for most proteins for which methyl resonance assignment is sought.

For the subset of the data for which a comparison to MAGIC was possible, MethylFLYA generated more strong assignments with higher accuracy (FLYA correct: 168, errors: 5; MAGIC correct: 130, errors: 50). The error rate of MAGIC on the two benchmark cases, EIN and ATCase, was ~10% when using the parameters that resulted in the highest MAGIC score (Supplementary Table 6, Supplementary Fig. 10). However, a significantly worse performance of MAGIC was found on the HSP90 data (Supplementary Fig. 10), which likely reflects a reduced quality of the methyl–methyl NOESY data (Supplementary Fig. 6). We note that the HSP90 dataset used in this study features Ile-$\delta1$, dimethyl Leu-$\delta1/2$ and Val-$\gamma1/2$ labeling, and, consequently, a significantly sparser methyl–methyl NOESY network (78 labeled methyl groups and 330 3D NOE peaks), when compared to the HSP90 dataset employed in the original MAGIC study (111 labeled methyl groups and 686 3D NOE peaks)[33]. The latter data were obtained on the N-terminal domain of HSP90 $^1H,^{13}C$-labeled on Ala-$\beta$, Met-$\epsilon$, Thr-$\gamma_2$, Ile-$\delta_1$, dimethyl Leu-$\delta_{1/2}$ and Val-$\gamma_{1/2}$ methyls, for which the authors report confident assignments of 88% of methyls with high accuracy (94%)[33].

A comparison of the assignments found by the different methods reveals that MAGMA and MethylFLYA produce the most similar solutions, which agree on 288 of the methyl assignments on this benchmark (Fig. 5, Supplementary Fig. 11, Supplementary Table 1). In contrast, MethylFLYA shares only 184 and 96 assignments with MAP-XSII and FLAMEnGO2.0, respectively. The intersection profiles are protein-specific (Supplementary Fig. 11), however, overall, a high degree of overlap with MethylFLYA solutions is seen for MAGMA (Supplementary Fig. 11, Supplementary Table 1). There are instances of confident assignments by MAGMA that are deemed tentative or ambiguous by MethylFLYA, and vice versa (Supplementary Table 1). Given that both protocols were given the same input data, a possible explanation for such assignment differences could be algorithm-specific parameters. The distance cutoffs used to generate the expected NOE contacts were similar for the two methods. Nonetheless, distance cutoff for MethylFLYA is applied as an $r^{-6}$ sum over the methyl proton distances, whereas MAGMA considers methyl carbon distances and, in addition, averages two methyl carbon positions for the geminal methyl groups of Leu and Val, which are treated separately by MethylFLYA. Therefore, the exact composition of the expected NOE contacts differs between the two methods, resulting in differences in restraint matching. Furthermore, MAGMA provides assignment results for one distance cutoff, whereas, for its confident assignments, MethylFLYA requires assignment consistency over three distance cutoff values separated by 0.5 Å (see Methods). Finally, MAGMA uses exact graph comparison algorithms to exhaustively sample

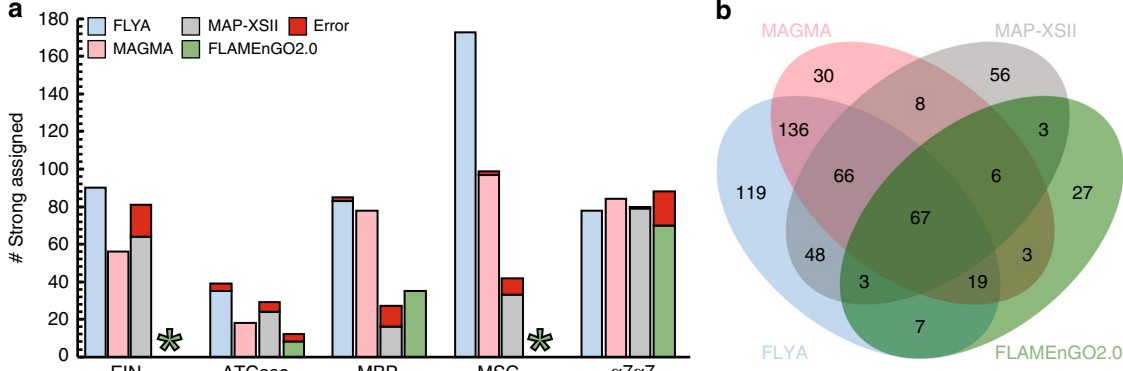

**Fig. 5** Comparison of MethylFLYA, MAGMA, MAP-XSII, and FLAMEnGO2.0. **a** The number of correctly and erroneously strongly (i.e. confidently) assigned methyl resonances for each of the cases is shown. Asterisks are given in the places where no confident (100%) assignments could be obtained with the FLAMEnGO2.0 software. **b** Mutual agreement of the methyl group resonance assignments among the four methods. Numbers of assigned methyls are given in the intersection sets

all assignment solutions that maximize the number of explained NOEs. In contrast, the evolutionary algorithm in MethylFLYA uses a heuristic to converge on a subset of most likely solutions, relying on differences between parallel runs of the algorithm to assess assignment self-consistency. Despite the listed differences, the high overlap in assignment solutions between MethylFLYA and MAGMA and their high accuracies demonstrate the complementarity of these two methods. Comparing the solutions from the two methods therefore constitutes a useful cross-validation approach, as the methyl assignments in the intersection of the two methods are completely accurate (Supplementary Table 1). Moreover, an agreement in erroneous assignment is rare across the tested methods, suggesting the utility of all existing protocols for assignment cross-validation (Supplementary Table 1).

In conclusion, we have presented an NOE-based approach to automatic methyl resonance assignment that is a significant advance over existing methods. Even though the general FLYA algorithm underlying MethylFLYA (Fig. 1) was originally designed to deal with through-bond, or a combination of through-bond and through-space information[34], the method proved powerful also for the assignment of methyl groups exclusively from NOESY and structural data (Fig. 2). This confirms earlier findings showing that FLYA is effective in assigning small proteins exclusively from $^{13}C$ and $^{15}N$-resolved NOESY data[36]. However, the assignment of methyl resonances in proteins as large as 360 kDa ($\alpha_7\alpha_7$), based on exclusively methyl–methyl NOEs, presents a considerably greater challenge because of data sparsity and minimal redundancy in data content. Nonetheless, MethylFLYA could generate as many, and in most cases significantly more, correct methyl assignments than existing algorithms (Fig. 5a). Only a very small number of assignments from MethylFLYA were erroneous, and all of these were to methyls spatially proximal to the correct assignment in the 3D structure (Supplementary Fig. 4), thus limiting their impact on studies relying on methyl assignments to deduce lower-resolution information, e.g., a protein-protein or protein-ligand interface. In other cases that rely strongly on site-specific interpretations of methyl resonance assignments, the user can apply stricter criteria on assignment confidence by requiring higher self-consistency of assignments from multiple parallel runs of the algorithm, e.g. from the default 80% to 90% or higher. Furthermore, the complementarity between MethylFLYA and MAGMA could be exploited. The user could also combine the automatic assignment with site-directed mutagenesis in regions of special interest. Any methyl resonance assignments previously known or newly acquired through site-directed mutagenesis can be fixed in the protocol of MethylFLYA, which will further aid its performance.

MethylFLYA is fast and robust in coping with ambiguous and erroneous NOEs, showing nearly identical performance on raw and refined NOESY data (Fig. 2, Table 1), and robustness to differences in input protein structures (Supplementary Fig. 9). MethylFLYA is also tolerant to ambiguity in the identity of Leu and Val resonances, whereas it significantly benefits from experimentally linking the methyl resonances from the geminal Leu/Val methyl groups (Fig. 3, Supplementary Fig. 7). The latter is further confirmed by the results of the MethylFLYA assignment of EIN using exclusively the 2D [$^1H$,$^{13}C$]-HMQC and the 4D methyl–methyl NOESY spectrum (Supplementary Fig. 7). We strongly advise running MethylFLYA with this information provided, which was also noted as critical and beneficial in the MAGMA[32] and MAGIC[33] studies, respectively. Stereospecific labeling of Leu and Val methyls is another promising approach[49] which is expected to further enhance the performance of MethylFLYA as it reduces the number of methyl resonances to be

assigned, removes the need for the geminal Leu/Val methyl pairing, and provides longer methyl–methyl NOE restraints.

We emphasize that MethylFLYA can provide reliable assignments for approximately a quarter of methyl resonances based solely on manually or CYPICK-derived peak lists from 2D [$^1H$,$^{13}C$]-HMQC and methyl–methyl NOESY spectra, and the "best guess" assignment of methyl residue types from, e.g., the BMRB chemical shift statistics. In such cases, we advise using a structure-based methyl chemical shift prediction[52] to identify any outliers of the characteristic residue type-based chemical shifts, which are likely to be assigned incorrectly based on BMRB statistics (Supplementary Fig. 8).

Given high-quality 2D [$^1H$,$^{13}C$]-HMQC and methyl–methyl NOESY spectra, automatic peak picking using CYPICK combined with MethylFLYA can provide assignment results of the same quality as the expert-prepared peak lists (Fig. 4, Supplementary Fig. 7). In fact, the combination with CYPICK can, in some cases, even lead to a higher number of strong assignments, or improved assignment accuracy (Fig. 4, Supplementary Fig. 7D, E). This finding is surprising considering that CYPICK peak lists show differences to the expert-generated ones and omit a fraction of the expert-picked signals, even for the highest quality spectra available (17% expert-picked peaks omitted for EIN, Supplementary Table 4). Given that successful applications of CYPICK with MethylFLYA are presently restricted to only two examples[50] (Fig. 4), wider applications of CYPICK with MethylFLYA will be required in the future to realistically judge its potential for fully-automated methyl resonance assignment.

A high fraction of overlap in confident methyl assignments between MAGMA and MethylFLYA indicates the complementarity of the two methods and can be useful in de novo assignment cross-validation (Fig. 5b). The utility of rapid, accurate methyl assignments is highlighted by recent studies that used NOEs between an unlabeled ligand and a methyl-labeled protein as restraints to generate models of the docked complex[32,38,53,54] and PCSs to measure reorientation of methyl groups upon ligand binding[55]. In the future, MethylFLYA could be extended to incorporate paramagnetic restraints, such as PREs or PCSs, or be combined with existing software packages that predominantly rely on these restraints[26,27]. Furthermore, MethylFLYA can straightforwardly be used to assign methyl resonances in solid-state NMR spectra[56].

## Methods

**Overview of the MethylFLYA algorithm.** The FLYA algorithm[34] determines resonance assignments by establishing an optimal mapping between expected peaks that are derived from knowledge of the protein sequence, types of NMR experiments, and, if available, 3D structure, and the observed peaks that are identified in the corresponding measured spectra. This mapping, and hence the assignments, are optimized by an evolutionary algorithm coupled to a local optimization routine[34,57]. MethylFLYA adopts the general FLYA algorithm for the assignment of methyl groups based on methyl–methyl NOEs and a known 3D structure. MethylFLYA uses the atom positions from the input protein structure and magnetization transfer pathways defined for each NMR experiment type to compute a network of expected peaks (Fig. 1). The mapping of expected peaks to measured ones starts from an initial population of random assignment solutions, which are optimized through successive generations by an evolutionary algorithm. To select the best individuals for recombination, a scoring function is employed, which takes into account the alignment of peaks assigned to the same atom, the completeness of the assignment, and the minimization of chemical shift degeneracy[34]. In each generation, a local optimization routine reassigns a subset of expected peaks through a defined number of iterations. This protocol is repeated multiple times starting from different random initial assignments. Details of the MethylFLYA algorithm are given in the following sections.

**MethylFLYA scripts.** Automated methyl assignment with MethylFLYA is performed by four scripts (CYANA macros written in the INCLAN[58] programming language) as described in Supplementary Methods. The initialization macro, init. cya, is executed when CYANA starts and reads the library of residues and NMR experiment types, as well as the protein sequence. The preparation macro, PREP.

cya, prepares the input data for the subsequent automated assignment calculations. This includes the splitting of experimental peak lists according to amino acid type (see below) and the setup for generating the corresponding expected peaks, which is saved in the expected peak list generation macro, peaklists.cya. PREP.cya may also include other preparatory steps, such as attaching hydrogen atoms to an input 3D structure from X-ray crystallography. The calculation macro, FLYA.cya, performs the actual automated assignment calculations using the peaklists.cya macro to generate the expected peaks with different values for the NOE distance cutoff (see below). After completion of the automated assignment calculations, the consolidation macro, CONSOL.cya, consolidates the assignment results from all individual optimization runs into a single consensus resonance assignment[34], which is the main result of MethylFLYA.

**Library of NMR experiments**. The types of NMR experiments that contribute input peak lists to MethylFLYA are defined in the CYANA library[34,36] (Fig. 1 and Supplementary Methods). For each spectrum type, a library entry defines the types of atoms that are observed in each spectral dimension and one or several magnetization transfer pathways that give rise to peaks. A magnetization transfer path is given by a probability for observing the corresponding experimental peak and a linear list of atom types that defines a molecular fragment, in which atoms must be of a given type (e.g. $^1H_{amide}$, $^1H_{aliphatic}$, $^1H_{aromatic}$, $^{13}C_{aliphatic}$, $^{13}C_{aromatic}$, $^{15}N$, etc.) and connected to the next atom in the list either by a covalent bond or by an NOE, i.e. a distance shorter than a given cutoff in the 3D structure. An expected peak is generated whenever a molecular fragment matches the covalent structure and, in case of NOEs, the 3D protein structure.

The following NMR experiments were used for MethylFLYA calculations in this paper: 2D [$^1H$,$^{13}C$]-HMQC (formally called C13HSQC in the CYANA library), 3D CCH-NOESY (CCNOESY3D; $^{13}C_1$, $^{13}C_2$, $^1H_2$ dimensions), 3D HCH-NOESY (C13NOESY; $^1H_1$, $^{13}C_2$, $^1H_2$ dimensions), 4D HCCH NOESY (CCNOESY; $^1H_1$, $^{13}C_1$, $^{13}C_2$, $^1H_2$ dimensions), and, optionally, 4D short-mixing time HCCH NOESY. The latter experiment can be recorded on a doubly methyl-labelled ([$^{13}C_{\delta1}^1H_3$/$^{13}C_{\delta2}^1H_3$]-Leu, [$^{13}C_{\gamma1}^1H_3$/$^{13}C_{\gamma2}^1H_3$]-Val) protein sample to correlate the geminal methyl groups of Leu and Val to each other. It is formally treated as an HCcCH-TOCSY experiment in the CYANA library for MethylFLYA. The experiment entries in the library are given in Supplementary Methods.

**Input peak lists**. MethylFLYA operates on peak lists with observed peaks from the measured NMR spectra that contribute data for the resonance assignment. The peak lists can be supplied in XEASY[59] format (Supplementary Methods), or other formats supported by CYANA. If residue type-specific information is available, e.g. from appropriately isotope-labeled samples, the [$^1H$,$^{13}C$]-HMQC peak list can be split into separate files containing only the methyl peaks of a certain residue type (called, for example, "C13HSQC_V.peaks" for Val peaks). The NOESY peak lists can be split similarly according to the two amino acid types involved in an NOE. In the MAGMA study, this information was available from manually assigned NOESY peak lists[32]. Here, unassigned NOESY peak lists are used as input, and each NOESY peak is automatically attributed to the amino acid types of the two [$^1H$,$^{13}C$]-HMQC peaks with the closest matching chemical shifts. Separate peak lists are written for each pair of amino acid types (called, for example, "CCNOESY_LL.peaks" and "CCNOESY_LV.peaks" for NOEs between two Leu residues or between Leu and Val, respectively). Splitting peak lists by residue types is optional. MethylFLYA also supports joint lists for the resonances of Leu/Val type, as well as for any other amino acid type combinations.

**Expected peak lists**. Lists of expected peaks are generated by MethylFLYA for a given set of experiments based on the protein sequence, the 3D structure, the library of NMR experiments, and the isotope labeling pattern. The input 3D structure file must contain hydrogen atoms. For all calculations in this paper, hydrogens were added to the input X-ray structures using the CYANA command 'atoms attach'. If residue type-specific experimental peak lists are available, MethylFLYA generates a separate expected [$^1H$,$^{13}C$]-HMQC peak list for each amino acid type and separate NOESY peak lists for each pair of amino acid types. Splitting the measured and expected peak lists by residue type(s) restricts the matching of expected peaks to measured peaks of the same amino acid type(s) in the automated assignment algorithm (Fig. 1).

The distance cutoff $d_{cut}$ for NOEs is an important parameter for generating expected NOESY cross peaks because the number of expected NOEs is approximately proportional to $d_{cut}^3$. MethylFLYA computes the effective distance for a pair of methyl groups as the $r^{-6}$-sum over the nine individual $^1H$-$^1H$ distances, i.e.

$$d_{eff} = \left( \sum_{i=1}^{3} \sum_{j=1}^{3} d_{ij}^{-6} \right)^{-1/6} \qquad (1)$$

where $d_{eff}$ stands for the effective distance, the sum includes all $^1H$ atoms of two methyl groups, and $d_{ij}$ is the Euclidean distance between individual methyl protons $i$ and $j$ that belong to two different methyl groups in the input structure. For the case that all $d_{ij}$ distances are assumed to be approximately equal, this yields $d_{eff} \approx 9^{-1/6} d_{ij} = 0.693 d_{ij}$. It should be noted that applying, for instance, a 5 Å cutoff to the

effective distance $d_{eff}$, allows inter-carbon distances between the two methyl groups of up to $5.0/0.693 + 2 \times 1.1 \approx 9.4$ Å, which includes twice the C–H bond length of 1.1 Å. To avoid giving high confidence to methyl assignments that are affected by minor changes of the NOE distance cutoff parameter $d_{cut}$, MethylFLYA performs assignment calculations with the three slightly different cutoffs of $d_{cut} - 0.5$ Å, $d_{cut}$, and $d_{cut} + 0.5$ Å, and determines the consensus assignments from the results obtained with the three cutoffs (see below).

For the calculations in this paper, the NOESY cross peak observation probability was optimized (see below) and then set to 0.1 for expected NOESY peaks, and to 1 for expected C13HSQC and short-mixing time NOESY (HCcCH) peaks for the calculations in this paper.

**Optimization of assignments**. Assignments are optimized by MethylFLYA using the same algorithm as the original FLYA method[34]. MethylFLYA uses chemical shift tolerances for the assignment calculations and results evaluation. These were set to 0.4 ppm for $^{13}C$ and 0.04 ppm for $^1H$ chemical shifts for all calculations of this paper. The population size for the evolutionary optimization algorithm[34] was set to 200, the value that was previously found to be optimal for exclusively NOESY-based FLYA calculations[36]. The number of iterations of the local optimization routine that is coupled to the evolutionary algorithm was kept at the default value of 15,000. For each distance cutoff value, MethylFLYA performs 100 independent runs of the optimization algorithm with identical input data and parameters that start from different initial random assignments.

**Consensus assignments**. It is important for an assignment algorithm to distinguish reliable assignments, in which the algorithm has a high confidence, from others that are tentative or ambiguous. To establish the confidence of the assignment of an individual atom, MethylFLYA analyzes the chemical shift values obtained in a series of independent runs of the optimization algorithm. The global maximum of the sum of Gaussians centered at the chemical shift values of the given atom in the individual optimization runs defines the consensus chemical shift value of the atom[34]. The standard deviation of these Gaussians is set to the chemical shift tolerance value of the atom (0.4 ppm for $^{13}C$ and 0.04 ppm for $^1H$). A consensus assignment is classified as "strong" (reliable) if more than 80% of the integral of the sum of Gaussians is concentrated in the region of the consensus chemical shift ± tolerance, i.e. if more than 80% of the individual runs yielded (within the tolerance) the same chemical shift value. It has been shown for the original FLYA algorithm that strong assignments are much more accurate than the remaining "weak" ones[34].

In MethylFLYA, consolidation into consensus assignments is enhanced in three ways over the original FLYA algorithm. (i) Three series of 100 individual runs are performed with three slightly different distance cutoffs for the generation of expected NOESY peaks (see above), and the consolidation is performed over all $3 \times 100$ individual runs of the optimization algorithm. This makes the algorithm less susceptible to the, necessarily somewhat arbitrary, choice of the NOE distance cutoff value, thereby reducing the number of erroneous strong assignments. (ii) Special measures are necessary for the geminal methyls of Leu and Val, for which the stereospecific assignment is unknown a priori. In this case, the chemical shift values obtained for the two methyls in the individual runs are redistributed such that the consensus assignments of the first/second methyl group are determined from the smaller/larger of the two chemical shift values in each run, and FLYA does not attempt to determine stereospecific assignments. In the original FLYA algorithm[34] this approach was applied independently to the $^1H$ pair and the $^{13}C$ pair of geminal Leu or Val methyl groups. This could result in inconsistent consensus assignments for the $^1H$ and $^{13}C$ resonances of Leu and Val geminal methyl groups, even though the underlying $^1H$ and $^{13}C$ assignments from the individual runs were always consistent with each other. To avoid this problem, the $^1H$ and $^{13}C$ chemical shifts of Leu and Val geminal methyl groups are consolidated jointly in MethylFLYA. (iii) Methyl assignments are only accepted as strong if at least one methyl–methyl NOE is assigned to the methyl group. This excludes assignments for which no experimental basis exists.

**MethylFLYA output**. At the end of an assignment run, MethylFLYA outputs the list of consensus chemical shifts (consol.prot) and a table with assignment results (consol.tab). In the consol.tab file, strong (reliable) assignments are marked with the label 'strong'. Other, tentative and ambiguous assignments are also reported for possible manual inspection. Further assignment statistics are given in the flya.txt file. It reports the number of expected, measured, and assigned peaks for each peak list, which are useful to detect problems with individual spectra or the assignment as a whole. In addition, more detailed information about the reliability of each resonance assignment is given, and, for each assignable atom, the expected and mapped measured peaks that have been used to establish its assignment are reported.

**Optimization of MethylFLYA parameters**. To establish optimal parameters for the MethylFLYA calculations, we tested a range of values for the methyl $^1H$–$^1H$ distance cutoffs for the generation of expected NOESY cross peaks, $d_{cut} = 3.0, 3.5, ..., 8.0$ Å (Supplementary Figs. 1, 2), observation probabilities for expected methyl–methyl NOESY peaks, $p_{NOE} = 0.1, 0.2, ..., 0.9$ (Supplementary Figs. 1, 2),

and the number of independent assignment optimization runs (Supplementary Fig. 3).

**Automated peak picking with CYPICK**. The CYPICK[37] algorithm for automated peak picking was applied to the NOESY spectra of EIN, ATCase, and HSP90. CYPICK relies on analyzing 2D contour lines of the spectrum, which are placed at intensity levels $I_i = \beta L \gamma^i$, where $i = 0, 1,\ldots$ and $L$ is the noise level of the spectrum that is estimated automatically by CYPICK. In this study, we used baseline factors $\beta = 2, 3, 4, 5, 10$ while keeping $\gamma$ fixed at 1.3. The scaling factors for the spectral dimensions[37] were set to 0.18 and 0.16 ppm for the first and second $^{13}C$ dimension, and 0.036 ppm for the $^1H$ dimension. The manually prepared or CYPICK-generated 2D [$^1H$,$^{13}C$]-HMQC peak lists were used as a frequency filter in CYPICK, restricting peak picking in the $^{13}C/^{13}C$-separated NOESY spectrum to locations within 0.01/0.1 ppm $^1H/^{13}C$ chemical shift from a [$^1H$,$^{13}C$]-HMQC peak position. Local maxima within the tolerance range that fulfilled the circularity and convexity criteria[37] were considered as peaks and stored in the peak list.

The peak picking performance was assessed by computing the find, artifact, and overall scores (with an artifact weight of 0.2) with respect to manually prepared reference peak lists[32] using a tolerance of 0.04 ppm for $^1H$ and 0.4 ppm for $^{13}C$ chemical shifts, as described in the CYPICK publication[37].

**MethylFLYA calculations using minimal input information**. The 2D HMQC spectrum was picked manually using Sparky software[42]. The BMRB chemical shift statistics[51] and the known number of Ile, Ala, Leu, Val residues in the protein sequence were used to generate the "best-guess" assignment of the peaks in the 2D $^1H$-$^{13}C$ HMQC spectrum to three methyl residue types: Ile, Ala, and Leu/Val (Supplementary Fig. 7A). The average methyl $^1H$ and $^{13}C$ chemical shifts ± one standard deviation were used to define regions associated to each methyl residue type. To assign residue types to peaks in the region of the spectrum where Ala and Leu/Val types overlap (Supplementary Fig. 7B), two strategies were followed: (i) the number of expected Ala peaks was maximized by attributing all the peaks in the overlapped regions to the Ala type; (ii) the peaks in the overlapped and border regions between the two types were assigned to both Ala and Leu/Val type (Supplementary Table 5). For the MethylFLYA calculations, the 4D methyl–methyl NOESY spectrum was reanalyzed based on the newly acquired 2D $^1H$-$^{13}C$ HMQC peak list. The NOESY peaks in the planes of overlapped resonances in the 2D spectrum were repicked. NOESY cross peaks were attributed automatically by CYANA to the closest methyl groups from the 2D [$^1H$,$^{13}C$]-HMQC spectrum (see above; section Input peak lists). SHIFTX2[52] was used to predict methyl resonances based on the EIN protein structure (PDB ID 1EZA) in order to identify any significant deviations from the BMRB statistics[51].

The Leu/Val geminal methyl group pairing was performed based on the reference assignment, restricted to the resolved methyl resonances in the Leu/Val region of the 2D [$^1H$,$^{13}C$]-HMQC spectrum (Supplementary Table 5 (iii)). If the geminal pair of any given Leu/Val resonance from the reference assignment mapped to an overlapped peak in the spectrum, both methyl resonances were removed from the geminal pairing. The geminal Leu/Val methyl pairs were supplied to MethylFLYA calculations in the form of an HCcCH peak list (see above; section Library of NMR experiments).

**Comparison with other assignment algorithms**. The performance of the alternative structure-based methyl assignment algorithms MAGMA[32], MAP-XSII[29], and FLAMEnGO2.0[31] has been compared earlier[32]. Here, we used the available results and identical parameters[32], with the exception of the MSG dataset, for which the calculations were repeated using the crystal structure (PDB ID 1D8C). A comparison to the MAGIC method was performed on a subset of the proteins for which NOESY spectra were available (EIN, ATCase, and HSP90). Both filtered and unfiltered manually picked methyl NOE peak lists were tested, as well as a range of distance thresholds (lower 4, 5, 6, 7 Å and higher 7, 8, 9, 10 Å, respectively; see Supplementary Table 6) for computing the inter-methyl connectivity network from the X-ray structure, and peak matching tolerance values (0.1, 0.2, 0.4 ppm for $^{13}C$; 0.01, 0.02, 0.04 ppm for $^1H$; see Supplementary Table 6). The mutual agreement between the resonance assignments generated by the different methods was visualized using an online tool available at the GPCRdb web interface (http://www.gpcrdb.org/signprot/statistics).

## Data availability

Experimental input data and corresponding MethylFLYA output data are available at http://www.cyana.org/methylflya.tgz. Other data are available from the corresponding author upon reasonable request.

## Code availability

MethylFLYA scripts for CYANA are available in http://www.cyana.org/methylflya.tgz.

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

## Acknowledgements

We thank Prof. Andrew Baldwin for help with the MAGMA benchmark data set, and Marta Carneiro, Eiso Ab and Gregg Siegal for providing the HSP90 data set. Financial support by a Eurostars grant of the Swiss Confederation and a Grant-in-Aid for Scientific Research of the Japan Society for the Promotion of Science (JSPS) is gratefully acknowledged.

## Author contributions

I.P. and P.G. designed and performed research, and wrote the paper. J.M.W. implemented and performed automated peak picking. T.R.A. assisted in spectral processing, analysis, and data interpretation. All authors contributed to data interpretation and commented on the paper.

## Competing interests

The authors declare no competing interests.
