## [Peer Review File · Nature Communications]

Reviewers' comments:

Reviewer #1 (Remarks to the Author):

This paper reports a new method (called methylFLYA) for automated NMR resonance assignments of isotope-labeled methyl groups. Manual assignment of methyl groups is a very time consuming albeit necessary step for investigation of large proteins or complexes by NMR. The presented method is largely based on the previously published FLYA algorithm incorporated in the CYANA software to match expected NOE peaks (from already known 3D structures) to experimentally measured CH₃-CH₃ NOEs. The efficiency of methylFLYA is compared to other available approaches (namely MAGMA, MAP-XSII and FLAMEnGO2.0) in term of performance, using the same input data. The reliability and efficiency of the approach is also tested on raw peak-lists, reduced input data sets and automatically picked peaks with CYPICK approach from the same group. methylFLYA almost consistently finds more correct assignment than other approaches with a very low error-rate. The method also conveniently reports on the robustness of the proposed assignments (confident, weak etc...)

The whole procedure is original (regarding methyl assignment), very efficient and potentially of great use for the NMR community since human intervention is greatly limited and the time to analyze the data is drastically reduced compared to manual other automated approaches commonly used by the community. The article is well written and the methods are exhaustively described.

Remarks:

- I regret the lack of comparison with the recent MAGIC approach but the authors explain why it wasn't possible technically. However it would still be interesting to see the results for the 2 proteins for which all data to run MAGIC were available.
- Table 1: The number of confidently assigned methyl resonances are listed for All, Correct and erroneous assignments. However, the numbers here do not add up. I imagined it is because among the methylFLYA assignments, some have no reference assignments or maybe those (neither correct nor erroneous) have ambiguous assignment? It should be made more clear.
- Page 6, section Assignment completeness and accuracy. The numbers of incorrectly assigned methyl groups reported for ATCase is 3 in the text. But if I count those in Table S1 and see the ones listed in table S2, there are 4 CH₃ groups for which the confident assignment was erroneous. Unless the authors consider the I42/I44 swaps as a single error...
- Page 13, in the discussion : 5 errors in total are mentioned for Methyl FLYA. As I just noted, it should be 6.
- If the assignments from all methods (or at least magna) are combined with methylFLYA, what would be the best achievable accuracy in terms of assignments? The author could extend their discussion on the complementarity of the different approaches.
- Related to that, can CYPICK be combined with MAGMA? Owing to the very low error-rate of MAGMA, it might be advantageous.
- Increasing the distance cutoff seems to help retrieving more correct assignments (while also generating more errors). Would it be possible to have a sort of 2-pass approach where methylFLYA is first run conservatively? Then running a 2nd pass, using previously found assignments and more generous parameters, such as larger *d_cut*, to try and recover more assignments?

Reviewer #2 (Remarks to the Author):

This manuscript extends the application range of the (at this time) quite general software FLYA. In the context of methyl resonance assignment, earlier attempts, including in particular also an earlier comparison, exist and the authors correctly refer to this (rather than repeating it again). A number of my questions were resolved when I arrived to the Methods section, which is at the end although logically this makes no sense (an example is the use of \pm signs together with dcut values, page 6 top). Still, some questions remain:

An interesting question (in my eyes anyway) are the consequences of errors that this, and likely any, automatic procedure introduces. The text points out that the five or so errors (e.g. Table 1) occur in places crowded with methyl groups (Fig. 1C), but what do these errors mean for a subsequent analysis (e.g. if the initial structure used is from a homologous protein, and a structure or binding study is intended)?

I couldn't quite match the numbers given in the lower part of page 13 with those in some figures (Figs. 4 and possibly S7).

I also had some more technical problems regarding the figure referencing. Firstly, figure 5 is mentioned first in the text; shouldn't it be figure 1? It is also mentioned on page 13 (towards the bottom) although I don't really understand why (is it Fig. 4?), and again once in the Discussion, and several times in Methods. The whole thing has to do with the placement of the Methods section. Some figure text is rather small (e.g. Fig. S1; the single point in the lower MSG part in this figure did raise my interest but this may be a bit picky).

The naming of the different methods is somewhat inconsistent. "FLAMEnGO2.0" is also referred to as simply "FLAME", and "MAX-XSII" as "MAX-XS" (e.g. Fig. 4). Another such name is HSP90 or HSP90-N. Maybe a clear definition regarding peak picking of "manual", "filtered" and "unfiltered" could be added (and "frequency-filter"), in particular since in some cases, "manual peak picking ... remains the best approach ... for MethylFLYA" (page 11).

Is Fig. S2 based on the same data as Fig. S1?

In summary, I liked to read the manuscript and consider it of rather general interest. Thus, after handling the above questions, I am in favor of its publication.

Reviewer #3 (Remarks to the Author):

The present work describes methylFLYA, a software aiming to obtain automated assignments of methyl groups using NOE distance information. The idea is not novel in NMR community, since it has been recognized many years now that is a viable approach to overcome laborious assignment strategies of large perdeuterated proteins. Of course, to tackle the problem, any software requires a reference structure. With these considerations in mind, the authors demonstrate that their software outperforms previous efforts on a test sample of five proteins that has been extensively studied and has been used as input data for software development.

Technically, the paper is good. The strong point is that it builds on many years of experience in related applications and on existing software architecture (FLYA). In my view, the distinctive feature of methylFLYA is the fact that it uses three different distance cut-offs in simulating NOE peaks from the given structures to match the experimental ones and consolidate the reported assignments. In principle, the software is a useful tool for the NMR community.

The problem is that methylFLYA, and perhaps the other software as well, depend on "certain underlying assumptions" to meet the performance standards presented here. These assumptions do not reflect on structural investigations based exclusively on the methyl-methyl NOESY spectrum. There is one detail that is not mentioned at all in the main text but only in Methods and it seems to be of paramount importance to automation and software performance. For the paper to be accepted, the authors would need to address properly the concerns raised below and it would be appropriate to change scope from complete automation to user-guided automation. This direction does not

undermine automation. Instead, users should be aware of the bottlenecks associated with it.

Major points:

1) The authors should be fair and state clearly the importance of the 2D ¹H-¹³C HMQC spectrum for automation. If the 2D peaks are used as anchors for clustering the peaks in 3D or 4D NOESY spectra or allow methylFLYA to deduce the number of methyl atoms in overlapping regions, the usability of methylFLYA (and the other software mentioned in the paper) becomes questionable. As seen in Figure S3 for the smallest target that consists of 116 methyl peaks there is substantial overlap. In my view the exact position of each peak cannot be discerned by the 2D HMQC alone, as it would be for any protein with no a priori information. Obviously overlap gets worse as the number of methyl frequencies increases. If "complete automation" is the key question here, then the 2D HMQC should be picked using CYPICK or any other picking software and compare the performance as to the situation where the 2D peak positions are known due to prior analysis. I encourage the authors to declare from the beginning that the optimization was performed by knowing peak positions in the 2D spectrum, (A)ILV identity of 2D peaks, and pairing for 2D LV peaks. As long the other software were provided the same information, then there is no doubt that methylFLYA performs better. The potential user, however, should be informed what to expect when only HMQC and NOESY spectra are available and possible workarounds that can enhance performance of methylFLYA as suggested below.

2) The use of automated peak picking is not justified by the results and certainly is not of practical use. It's hard to imagine a user that would not curate manually the peaks picked by any software. The authors did perform manual analysis themselves to prepare the unfiltered NOESY peaklists. However, they state that: "Manual analysis of NOESY data is a time-consuming and inherently user-biased task, complicated by spectral artifacts, low signal-to-noise ratios, and signal overlaps (Fig. S6)" to finally conclude that "manual peak picking of the NOESY spectra remains the best approach for preparing the input data for MethylFLYA." In fact, when dealing with a single methyl-methyl NOESY spectrum the user can, and should, go fast through all strips or planes and with minimum effort provide methylFLYA important information. Instead of aiming for complete automation, I strongly suggest to apply human reasoning and guide automation. 4D NOESY spectra offer tremendous possibilities. The authors should test the following workflow to discern overlap and pairing of LV peaks for EIN protein. Pick the 2D HMQC and curate if required. For every resolved L or V methyl peak inspect the plane of the 4D NOESY and curate as you operate. By virtue of distance dependence, the stronger cross-peak will correspond to the geminal methyl of the particular L or V. The advantage is twofold. The user can place anchors (peak positions) in overlapping regions of the 2D HMQC and at the same time can define the pairing. Once the NOE patterns are understood he could also operate in overlapping regions to identify more anchor points and pairs of LV peaks. Run methylFLYA and report the results. My expectation is that performance will be close to the "dark grey" scenario of Fig. 2, though the overlap in A and I methyl groups should have a negative impact on the overall performance. This would be the true value of automation. The human brain cannot decipher the intricate NOE network but the software requires human guidance to succeed without a priori information.

3) The paper needs restructuring. Fig. 5 should come first explaining in detail the input data. In fact, is cited first in the introduction. CYPICK section doesn't offer much and could be much shorter. A last section in Results should be added and discuss real case applications using 2D HMQC and 4D NOESY of EIN protein as an example reporting methylFLYA performance for complete automation (point 1) and user-guided automation (point 2). It should also explain the user the practical aspects that would assist him to master the software for optimum results, e.g. preparation of the input peaklists, post analysis tips, if any, to confirm and restrict assignments in following rounds, etc.

Minor points:

1) What would be the effect of 5 cut-off values instead of 3 in consolidating the assignments, e.g. 5.5 +/- 1 Angstrom for all proteins?

2) Why HSP90 is not part of the benchmark set but is mentioned only in the CYPICK section?

3) If technically not difficult, would be interesting to know MAGIC performance for EIN, ATCase, and HSP90.

4) ^{13}C and ^1H frequencies are counted independently throughout the text for the statistics. The correct assignments however are only the ones where both C and H frequencies are correct because in 4D NOESY spectra the frequencies are correlated. From Table S1, MBP protein:

8 VAL CG1 20.962 0.876 (!)

If an assignment decision is made then both frequencies should be present, it makes no sense that only one frequency is derived from the 4D NOESY spectrum.

139 LEU CD1 25.914 0.787 =!

If proton is wrong then both assignments are wrong because they refer to a different C-H peak. In a follow-up titration experiment, if this C-H peak is affected, will report a false methyl group.

5) Perhaps the authors could explain why the best results are obtained with low probability for expected NOESY peaks. Or what is the importance of this parameter. What would be the result if set close to zero, e.g. 0.001?

In response to the comments we have prepared a revised version of the manuscript that takes into account the points raised by the reviewers, as follows.

Reviewer #1 (Remarks to the Author):

I regret the comparison the lack of comparison with the recent MAGIC approach but the authors explain why it wasn't possible technically. However it would still be interested to see the results for the 2 proteins for which all data to run MAGIC were available.

We now include a comparison with the MAGIC protocol for three proteins, ATCase, EIN, and HSP90, for which we have access to the methyl-methyl NOESY spectra and could therefore obtain NOE signal intensities. The results are summarized in Table 1, Supplementary Table 6 and Supplementary Fig. 9, and discussed in the text on pp. 18/19. We ran the protocol on both unfiltered NOESY peak lists and, where available, the filtered peak lists (see Methods). In addition, we tested the performance over a range of distance thresholds and peak matching tolerances. Changing these parameters from defaults generally led to a decrease in performance (Supplementary Table 6). Please note that increasing the peak matching tolerance had a major impact on the calculation time. With $^{13}\text{C}/^1\text{H}$ tolerances of 0.4/0.04 ppm, the MAGIC calculation for EIN did not complete within 10 days (based on the log file, about 30% had been done).

Table 1: The number of confidently assigned methyl resonances are listed for All, Correct and erroneous assignments. However, the numbers here do not add up. I imagined it is because among the methylFLYA assignments, some have no reference assignments or maybe those (neither correct nor erroneous) have ambiguous assignment ? It should be made more clear

The situation is as suggested by the reviewer. We thank the reviewer for pointing out this source of confusion. The difference between “All” and the sum of “Correct” and “Erroneous” indeed captures the confident (strong) FLYA assignments for which a reference assignment is missing. We added a footnote to Table 1 clarify this point. In addition, we now always report the number of assignments per ^1H - ^{13}C pair rather than for the ^1H , ^{13}C resonances separately.

Page 6, section Assignment completeness and accuracy. The numbers of incorrectly assigned methyl groups reported for ATCase is 3 in the text. But if I count those in Table S1 and see the ones listed in table S2, there are 4 CH3 groups for which the confident assignment was erroneous. Unless the authors consider the I42/I44 swaps a s a single error...

We thank the reviewer for pointing out this inconsistency. The number in the text was wrong and has been corrected.

Page 13, in the discussion : 5 errors in total are mentioned for Methyl FLYA. As I just noted, it should be 6.

We have corrected this mistake in the text.

If the assignments from all methods (or at least magma) are combined with methylFLYA, what would be the best achievable accuracy in terms of assignments ? The author could extend their discussion on the complementarity of the different approaches.

Indeed, the assignment accuracy for the intersection between MAGMA and FLYA is complete. We thank the reviewer for this insight and have expanded our discussion on this topic in the text on pp. 20 and 21.

Related to that, can CYPICK be combined with MAGMA? Owing to the very low error-rate of MAGMA, it might be advantageous.

This is a very good suggestion. However, MAGMA requires that NOE peaks are explicitly assigned to their respective methyls from the HMQC spectrum and that a 2D contact list is provided as input. Given the large ambiguity in the assignment of automatically picked NOEs, which is currently not supported by MAGMA, the step would require expert filtering and unambiguous assignment of the NOESY peak list. For this reason, and because MAGMA is not the central subject of this paper, we did not provide such a comparison.

*Increasing the distance cutoff seems to help retrieving more correct assignments (while also generating more errors). Would it be possible to have a sort of **2-pass approach** where methylFLYA is first run conservatively? Then running a 2nd pass, using previously found assignments and more generous parameters, such as larger d_{cut} , to try and recover more assignments?*

This is also a very interesting suggestion. However, trials did so far not yield significant improvements. For the present paper, we therefore prefer to avoid additional complexity in the method that is not clearly beneficial for the results.

Reviewer #2 (*Remarks to the Author*):

This manuscript extends the application range of the (at this time) quite general software FLYA. In the context of methyl resonance assignment, earlier attempts, including in particular also an earlier comparison, exist and the authors correctly refer to this (rather than repeating it again). A number of my questions were resolved when I arrived to the Methods section, which is at the end although logically this makes no sense (an example is the use of \pm signs together with d_{cut} values, page 6 top).

We thank the reviewer for raising this point. We now introduce more explanation of the method early in the text to ensure a better flow of information in the manuscript and a better linkage between the Methods section and the rest of the manuscript. In particular, we moved the section 'Experimental data' from Methods up to the beginning of the Results as 'Benchmark data' on p. 6, we clearly explain the use of three distance cutoffs spaced by 0.5 Å and the filtered/unfiltered peak lists on p. 7.

Still, some questions remain:

- An interesting question (in my eyes anyway) are the consequences of errors that this, and likely any, automatic procedure introduces. The text points out that the five or so errors (e.g. Table 1) occur in places crowded with methyl groups (Fig. 1C), but what do these errors mean for a subsequent analysis (e.g. if the initial structure used is from a homologous protein, and a structure or binding study is intended)?

We thank the reviewer for this insight. We now comment on the potential impact of such assignment errors in Results on p. 8 and in the Discussion on p. 20.

I couldn't quite match the numbers given in the lower part of page 13 with those in some figures (Figs. 4 and possibly S7).

We thank the reviewer for pointing out the confusing reference to the result. The problem arose from reporting ^1H and ^{13}C assignments separately in the figures, while referring to methyl groups (^1H - ^{13}C pairs) in the text. As already mentioned in response to Reviewer 1, we now report throughout the paper all results per methyl group to avoid confusion.

I also had some more technical problems regarding the figure referencing. Firstly, figure 5 is mentioned first in the text; shouldn't it be figure 1? It is also mentioned on page 13 (towards the bottom) although I don't really understand why (is it Fig. 4?), and again once in the Discussion, and several times in Methods. The whole thing has to do with the placement of the Methods section.

We thank the reviewer for pointing out the confusing order of figures and mistakes in figure referencing. We switched also Figures 1 and 5 and fixed the incorrect references to the figure numbers. Because of the formatting requirement of the journal, we keep the Methods section at the end. But, as mentioned above, we adapted the text to improve the references to the Methods section and introduce information necessary for understanding of the results at appropriate places earlier in the text, i.e. at the start of the Results section.

Some figure text is rather small (e.g. Fig. S1; the single point in the lower MSG part in this figure did raise my interest but this may be a bit picky).

It is correct that labeling in some figures in the Supporting Information is small (e.g. Supplementary Figs. 1 and 7). We wanted to report a large amount of detail for reference here (that does not have to be studied in detail to understand the results in the main part of the paper). We have enlarged some of the corresponding figures somewhat. If desired by the Editor, the information could also be split into several pages.

We have double-checked the outlier for MSG in Supplementary Fig. 1, confirming that the value is correct.

The naming of the different methods is somewhat inconsistent. "FLAMEnGO2.0" is also referred to as simply "FLAME", and "MAX-XSII" as "MAX-XS" (e.g. Fig. 4). Another such name is HSP90 or HSP90-N.

We have fixed these inconsistencies.

Maybe a clear definition regarding peak picking of "manual", "filtered" and "unfiltered" could be added (and "frequency-filter"), in particular since in some cases, "manual peak picking ... remains the best approach ... for MethylFLYA" (page 11).

We thank the reviewer for reading our manuscript carefully and pointing out the need for clarifying and defining these terms. We now introduced these with the first mention of these terms in the text, i.e. on pp. 6/7 for filtered/unfiltered peak lists and on p. 27 for the frequency filter.

Is Fig. S2 based on the same data as Fig. S1?

This is indeed the case. We now clearly indicate this in the legend of Supplementary Fig. 2. We included Supplementary Fig. 2, in which the parameter specifying the NOE probability is fixed in each column, to more clearly illustrate the effect of varying the distance threshold parameter for a given, fixed NOE probability value.

Reviewer #3 (Remarks to the Author):

For the paper to be accepted, the authors would need to address properly the concerns raised below and it would be appropriate to change scope from complete automation to user-guided automation. This direction does not undermine automation. Instead, users should be aware of the bottlenecks associated with it.

We agree with the reviewer and have adapted the wording in the manuscript and added a new section to Results that discusses the proposed user-guided automation (pp. 16-17).

1) The authors should be fair and state clearly the importance of the 2D 1H-13C HMQC spectrum for automation. If the 2D peaks are used as anchors for clustering the peaks in 3D or 4D NOESY spectra or allow methylFLYA to deduce the number of methyl atoms in overlapping regions, the usability of methylFLYA (and the other software mentioned in the paper) becomes questionable.

In contrast to the other algorithms, MethylFLYA does not mandatorily require the 2D peaks as anchors for the NOESY peaks, as it uses as input unassigned methyl-methyl NOESY peak lists without requiring the use of 2D peaks as anchors. However, we agree that when analyzing the methyl-methyl NOESY spectrum itself, the knowledge of the exact (reference) positions from the 2D 1H-13C HMQC spectrum provides additional information to the process, e.g. by revealing the number of methyls in overlapped regions. We now explicitly comment on this in the manuscript on p. 15 and proceed to reanalyze the spectrum and run MethylFLYA blind to such information as detailed in the new Results section 'MethylFLYA using minimal input information' on pp. 15–17.

As seen in Figure S3 for the smallest target that consists of 116 methyl peaks there is substantial overlap. In my view the exact position of each peak cannot be discerned by the 2D HMQC alone, as it would be for any protein with no a priori information. Obviously overlap gets worse as the number of methyl frequencies increases. If "complete automation" is the key question here, then the 2D HMQC should be picked using CYPICK or any other picking software and compare the performance as to the situation where the 2D peak positions are known due to prior analysis.

We thank the reviewer for raising these important points. We performed the proposed analysis both using CYPICK and manual peak picking and documented the strategy, new results, and valuable insight that came from these remarks in the new text on pp. 15–17, and the new Supplementary Table 5 and Supplementary Fig. 7.

I encourage the authors to declare from the beginning that the optimization was performed by knowing peak positions in the 2D spectrum, (A)ILV identity of 2D peaks, and pairing for 2D LV peaks. As long the other software were provided the same information, then there is no doubt that methylFLYA performs better.

We now explicitly declare this and discuss this in detail in the context of the new section in Results on pp. 15–17. We note that indeed the exactly same information is required and was provided to all other software (MAGMA, FLAMEnGO2.0, MAGIC, and MAP-XSII). The only exception to this rule was MAP-XSII software that does not support input of 2D LV pairing information, as was already noted before in the MAGMA study.

The potential user, however, should be informed what to expect when only HMQC and NOESY spectra are available and possible workarounds that can enhance performance of methylFLYA as suggested below.

We thank the reviewer for this suggestion, based on which we expanded the manuscript to include a stepwise user guided analysis of spectra starting from the completely “blind to the reference” scenario and subsequently increasing amount of information introduced to the FLYA software solely for the well-resolved peaks. We discuss these aspects in the new section in Results on pp. 15–17, Supplementary Table 5, and Supplementary Fig. 7, as already mentioned above.

2) The use of automated peak picking is not justified by the results and certainly is not of practical use. It's hard to imagine a user that would not curate manually the peaks picked by any software. The authors did perform manual analysis themselves to prepare the unfiltered NOESY peaklists. However, they state that: “Manual analysis of NOESY data is a time-consuming and inherently user-biased task, complicated by spectral artifacts, low signal-to-noise ratios, and signal overlaps (Fig. S6)” to finally conclude that “manual peak picking of the NOESY spectra remains the best approach for preparing the input data for MethylFLYA.”

We now removed such confusing statements from the manuscript and adjusted the text to reduce the scope originally presented based on the CYPICK data and instead focus more on the proposed user-guided protocol.

In fact, when dealing with a single methyl-methyl NOESY spectrum the user can, and should, go fast through all strips or planes and with minimum effort provide methylFLYA important information.

We thank the reviewer for this constructive suggestion, based on which we amended and expanded the manuscript. We reanalyzed the NOESY spectrum as proposed, which resulted in higher number of picked methyl-methyl NOEs compared to the

Instead of aiming for complete automation, I strongly suggest to apply human reasoning and guide automation. 4D NOESY spectra offer tremendous possibilities. The authors should test the following workflow to discern overlap and pairing of LV peaks for EIN protein. Pick the 2D HMQC and curate if required.

We followed the protocol as suggested and first picked manually the 2D HMQC spectrum.

For every resolved L or V methyl peak inspect the plane of the 4D NOESY and curate as you operate. By virtue of distance dependence, the stronger cross-peak will correspond to the geminal methyl of the particular L or V. The advantage is twofold. The user can place anchors (peak positions) in overlapping regions of the 2D HMQC and at the same time can define the pairing. Once the NOE patterns are understood he could also operate in overlapping regions to identify more anchor points and pairs of LV peaks.

This step was unfortunately not possible given the available data as the 4D NOESY spectrum was recorded on a [$^{13}\text{C}\text{H}_3$, $^{12}\text{C}\text{D}_3$]-Leu,Val labeled sample in which only one of the geminal methyl groups of each Leu/Val residue is labelled. As such, the cross-peak from the geminal methyls was not available in the planes of the 4D NOESY spectrum. We instead imagined a scenario in which a [$^{13}\text{C}\delta_1/\gamma_1\text{H}_3$, $^{13}\text{C}\delta_2/\gamma_2\text{H}_3$]-Leu/Val labeled sample was prepared allowing for a short-mixing time methyl-methyl NOESY experiment to be performed. However, pairing of the geminal Leu/Val methyl resonances was restricted to only the non-overlapping peaks to avoid the assumption that the user could unambiguously match up all of the geminal pairs based on those NOESY data. We described the approach in detail in the main text on pp. 15–17 and the new Supplementary Table 5 and Supplementary Fig. 7.

Run methylFLYA and report the results. My expectation is that performance will be close to the “dark grey” scenario of Fig. 2, though the overlap in A and I methyl groups should have a negative impact on the overall performance.

All new results generated following the proposed protocol are reported.

This would be the true value of automation. The human brain cannot decipher the intricate NOE network but the software requires human guidance to succeed without a priori information.

We agree and thank the reviewer for all of the constructive suggestions.

3) The paper needs restructuring. Fig. 5 should come first explaining in detail the input data. In fact, is cited first in the introduction.

We rearranged the figures by exchanging Figs. 1 and 5, as proposed.

CYPICK section doesn't offer much and could be much shorter.

We shortened this section, as advised.

A last section in Results should be added and discuss real case applications using 2D HMQC and 4D NOESY of EIN protein as an example reporting methylFLYA performance for complete automation (point 1) and user-guided automation (point 2). It should also explain the user the practical aspects that would assist him to master the software for optimum results, e.g. preparation of the input peaklists, post analysis tips, if any, to confirm and restrict assignments in following rounds, etc.

These new results and discussion of the requirements and preparation have been added, as mentioned above.

Minor points:

1) What would be the effect of 5 cut-off values instead of 3 in consolidating the assignments, e.g. 5.5 +/- 1 Angstrom for all proteins?

During the development of MethylFLYA, we tested this and other combinations of distance cutoffs and chemical shift consolidation parameters, but could not see significant improvements. To avoid further, potentially distracting complexity in paper, we therefore prefer to keep the description to the recommended method. However, users can easily modify and run the protocol over as many distance threshold as desired and run consolidation accordingly.

2) Why HSP90 is not part of the benchmark set but is mentioned only in the CYPICK section?

We kept HSP90 separate, as it was also not part of the benchmark in the earlier MAGMA study.

3) If technically not difficult, would be interesting to know MAGIC performance for EIN, ATCase, and HSP90.

We performed these calculations with MAGIC. The results are summarized in Table 1, Supplementary Table 6 and Supplementary Fig. 9, and discussed in the text on pp. 18/19. See response to Reviewer 1 above.

4) 13C and 1H frequencies are counted independently throughout the text for the statistics. The correct assignments however are only the ones where both C and H frequencies are correct because in 4D NOESY spectra the frequencies are correlated. From Table S1, MBP protein:

8 VAL CG1 20.962 0.876 (!)

If an assignment decision is made then both frequencies should be present, it makes no sense that only one frequency is derived from the 4D NOESY spectrum.

139 LEU CD1 25.914 0.787 =!

If proton is wrong then both assignments are wrong because they refer to a different C-H peak. In a follow-up titration experiment, if this C-H peak is affected, will report a false methyl group.

We agree with this the reviewer and have changed this throughout the manuscript, as also requested by the other reviewers above.

5) Perhaps the authors could explain why the best results are obtained with low probability for expected NOESY peaks. Or what is the importance of this parameter. What would be the result if set close to zero, e.g. 0.001?

The results do not strongly depend on this observation probability. In the general FLYA algorithm, it provides a way to specify the relative importance of assigning peaks in a given spectrum relative to those in another spectrum. However, in the case of MethylFLYA, the essential data for the assignment comes all from the single type of spectrum (NOESY), such that the observation probability value have a smaller influence than in FLYA applications with different spectrum types. For instance, decreasing the observation probability parameter from 0.1 to 0.001 for the EIN data set results in a 6% decrease of the number of strong assignments, a 2% decrease of the correct assignments (the strong assignments with available reference assignment for cross-validation), and no assignment errors in both cases.

REVIEWERS' COMMENTS:

Reviewer #2 (Remarks to the Author):

Looking at the revised manuscript and in particular at the rebuttal, I feel that the authors have satisfactorily addressed the points raised. I have no further questions or comments.

Reviewer #3 (Remarks to the Author):

The manuscript is considerably improved as my previous concerns have been largely addressed. I thus recommend publication. In my view, SupFig7 is the most interesting part of the paper and should be a main figure.